# Advanced Prediction for Cyclic Bending Behavior of RC Columns Based on the Idealization of Reinforcement of Bond Properties

Peilun Shao, Gakuho Watanabe *  and Elfrido Elias Tita

Department of Civil and Environmental Engineering, Yamaguchi University, 2-16-1, Yamaguchi 7558611, Japan; a504wdu@yamaguchi-u.ac.jp (P.S.); neuz2012@gmail.com (E.E.T.)
* Correspondence: gakuho.w@yamaguchi-u.ac.jp

**Abstract:** The bonding characteristics between steel bars and concrete in reinforced concrete (RC) structures are crucial for the prediction of load-bearing capacity for seismic design. Nevertheless, most previous studies on bond-slip performance focus on the bond strength based on the pull-out experiments, it is often overlooked that the effect on the failure modes of RC members and the deformation performance due to the bond characteristics. In this research, the effect of the diameter and its arrangement of the reinforcement of the RC column on the bond failure mode and load-bearing capacity based on the cyclic loading tests and the FE analysis are carried out. In the cyclic loading test, it was found that two RC columns with different diameters and reinforcement arrangements showed distinct load-bearing capacity, deformation performance, and failure mode. Despite those columns having the same longitudinal reinforcement ratios. In addition, by applying an advanced finite element analysis using a bond-slip model that induces splitting failure, we succeeded in reproducing the cyclic deformation behavior and local damage obtained in experiments with high accuracy. The proposed model brings in the advanced prediction of the seismic behavior of RC structures and the enhancement of seismic resistance of social infrastructure facilities to earthquake disasters.

**Keywords:** cycle loading test; RC column; pinching effect; finite element analysis; bond-slip; pull-out; splitting

## 1. Introduction

Reinforced concrete (RC) columns are widely employed in the construction of buildings and infrastructures. Specifically, reinforced concrete piers are favored in projects such as highway, mountain, and cross-sea bridges due to their cost efficiency, ease of construction, exceptional durability, and resistance to seismic and corrosive forces. In the design and construction of RC columns, particularly piers, the bond performance between reinforcement and concrete is a critical consideration [1–3]. Adequate bond strength is essential in all reinforced concrete structures to ensure effective load transfer between concrete and reinforcement, satisfying strength and serviceability requirements. Most reinforced concrete structures experience degradations due to weakened bond strength at crucial locations (e.g., nodes at column bases), leading to reinforcement slippage and reduced load-bearing capacity and stiffness [4,5]. Bond behavior significantly affects fatigue and seismic performance [6]. Bond failure can result in stress redistribution, increased local deformation, or premature structural failure. Consequently, bond performance directly influences the load-bearing capacity, mechanical properties, and overall safety of reinforced concrete structures.

Furthermore, prior studies have shown that rigid body rotation occurs in response to longitudinal reinforcement extension. When the longitudinal reinforcement pulls out from the footing due to bond-slip failure [7], the rotational deformation (rocking) of RC columns becomes more pronounced than the bending deformation, substantially impacting

RC columns' strength, stiffness, and durability. Depending on the design strategy, RC columns that permit rotational deformation can absorb more significant deformation and energy, flexibly responding to large forces generated during earthquakes. In contrast, when rocking deformation is restricted, RC columns may become stiffer, however, RC columns are more prone to experience rapid failure progression once failure commences [8,9].

In summary, a proper evaluation of bond-slip performance in RC piers is crucial for seismic design. The bond performance between reinforcement and concrete is influenced by various factors [10], including concrete strength, hoop conditions, and the diameters and arrangements of longitudinal bars within the RC column [11–13]. The bond between reinforcement and concrete may exhibit different failure types depending on the diameters and arrangements of the longitudinal bars [14], primarily categorized into pull-out and splitting types of failure.

Ertzibengoa et al. [15] conducted numerous rebar pull-out experiments and effectively studied the cracks formed around the reinforcement due to annular destruction through the injection of fluorescent epoxy resin. However, their primary subject was flat steel rebar, which has a relatively narrow application scope in the actual bridge construction field compared to the widely used round reinforcements. Chapman et al. [16] also carried out a large number of pull-out experiments, specifically for different embedment lengths. However, the experiments involved early age concrete and smooth steel rebar, which do not have a wide range of applications in the actual bridge construction field. Khaksefidi et al. [17] employed ultra-high-strength concrete and used both high-strength and normal-strength reinforcement, conducting 60 rebar pull-out experiments with three embedment lengths to investigate the impact of concrete strength, bond length, rebar yield strength, and rebar geometry on the bond-slip failure mode. However, this research was conducted under the condition of absolute constraint on all sides and the monotonic pull-out experiment of a single rebar, which greatly overestimates the lateral constraint compared to an actual Reinforced Concrete (RC) bridge pier affected by an earthquake. The effect of cyclic loading was also not fully considered. Zhang et al. [18] conducted numerous rebar pull-out experiments to investigate the degradation of bonding performance caused by fatigue loading, especially for specimens with thin concrete cover layers. They applied repeated loads of different amplitudes and periods. However, this study did not consider the degradation of bond performance and the change of annular failure mode caused by the dense arrangement of a large number of longitudinal reinforcements in actual RC bridge structures.

In summary, mere rebar pull-out experiments cannot perfectly reproduce the bond-slip phenomenon of internal longitudinal rebars in RC bridge piers during an earthquake. We need to reproduce this through hysteresis load tests on actual RC columns, examining rebar strain and concrete damage, and finite element analysis. This will allow us to better judge the actual bond-slip phenomenon inside RC bridge piers during earthquakes, and thus more accurately predict the deformation performance of structures.

## 2. Experimental Programs

A significant number of cyclic loading tests have been conducted on reduced-scale RC columns to explore the seismic behavior of single-column RC bridge piers. The cross-sectional dimensions and reinforcement configurations of bridge piers differ based on the design codes of various countries. In Japan, it is common to determine the cross-sectional dimensions based on the earthquake resistance capacity method specified in the 1996 Road Bridge Specifications [19]. Hayakawa, R., Kawashima, K., and Watanabe, G. elucidated RC columns' bending resistance and deformation performance under 5 different loading protocol (unidirectional load, diagonal load, rectangular load, circular load, and elliptical load) square RC columns with a cross-section of 400 mm × 400 mm and D13 reinforcements [20]. Subsequently, to investigate the effects of thin-wall reinforcement, Shao, P., Watanabe, G., and Kosa, K. designed two specimens: an unreinforced RC specimen and an Ultra-thin reinforced specimen, both using D10 rebar under the cyclic loading test [21].

In this study, by comparing the experimental results of two base experimental bodies having different diameters and reinforcement bar arrangements before and after, aiming to comprehensively examine bond-slip behavior in RC columns during earthquakes. The deformation load capacity and reinforcement damage were compared with a reference experiment to assess the effect of reinforcement diameter and arrangement on bond-slip performance.

### 2.1. Design of Specimen with Densely Arranged, Small Diameter Reinforcements

In this study, we investigate the deformation performance and failure conditions using two RC specimens with different arrangements of steel bars but same longitudinal reinforcement ratios under cyclic loading, as illustrated in Figure 1. The experiments employed different loading devices, resulting in effective heights (measured from the bottom of the pier to the horizontal force point) of 1350 mm and 1600 mm for the two RC column specimens, and overall column heights of 1750 mm and 2565 mm. Both specimens had a square cross-section of 400 mm × 400 mm and footings with dimensions of 700 mm in height, 1300 mm in width, and 1100 mm in depth.

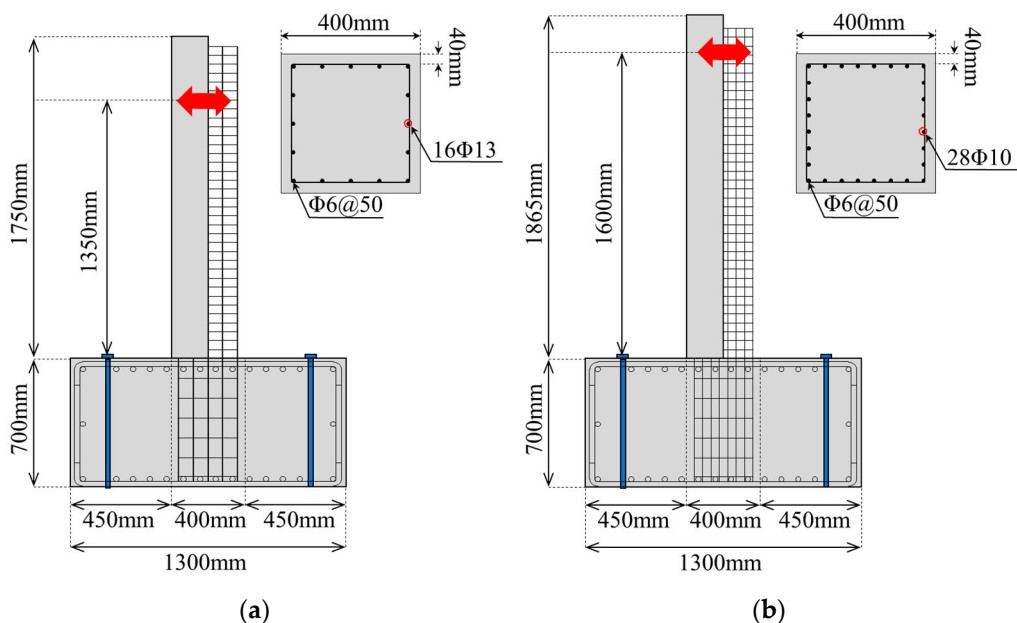

**Figure 1.** Reinforcement details of the RC columns specimens (units: mm). (**a**) Case-1. (**b**) Case-2.

As illustrated in Figure 1a, the RC column specimen utilized SD295A reinforcement bars, featuring 16 D13 bars as longitudinal reinforcement and D6 bars as hoop reinforcement, spaced at 50 mm intervals from the column base. The longitudinal reinforcement ratio and hoop reinforcement volume ratio for this case, referred to as Case-1, was 1.27% and 0.79%.

In contrast, Case-2, depicted in Figure 1b, involved an RC column specimen with the same longitudinal reinforcement ratio as Case-1 but incorporated thinner D10 deformed reinforcement bars in a denser arrangement. This design used 28 D10 SD295A reinforcement bars, maintaining a constant 40 mm cover concrete thickness and 50 mm intervals between hoop reinforcement in the vertical direction. The longitudinal reinforcement ratio and hoop reinforcement volume ratio for Case-2 were approximately the same as in Case-1, at 1.25% and 0.79%.

$$I_{rc} = I_c + \left(\frac{E_s}{E_c} - 1\right)\sum_{i=1}^{n}\left(I_{si} + A_i y_i^2\right) \tag{1}$$

In the study of the bending resistance characteristics of RC columns, we utilized Equation (1) to calculate the moments of inertia for the cross-sections of two RC columns

with different rebar diameters and arrangements. Case-1, which utilized large-diameter and low-density arrangement, resulted in a moment of inertia of $3.11 \times 10^9$ mm$^4$. Conversely, Case-2, which used small-diameter and high-density arrangement, had a moment of inertia of $3.13 \times 10^9$ mm$^4$. Not only were the reinforcement ratios essentially identical for both cases, but their moments of inertia were also roughly on par with each other.

In summary, Table 1 presents the specific parameters for each experimental subject. The concrete used was standard Portland cement with an aggregate maximum particle size of 20 mm. Furthermore, as per the Road Bridge Specifications [19], the elastic moduli for the longitudinal reinforcements and stirrups were set at 210 Gpa, while that of the concrete was 28.0 Gpa.

**Table 1.** Summary of Cyclic Loading Test Specimens.

| | **Strength Parameters (Mpa)** | | | | | | | | |
|---|---|---|---|---|---|---|---|---|---|
| | $f_c$ | $f_y{'}$ | $f_u{'}$ | $f_{hy}{'}$ | $f_{hu}{'}$ | $D_{lr}$ | $\rho$ | $\rho_w$ | $I_{rc}$ |
| Case-1 | 30.1 | 357 | 486 | 321 | 492 | D13 (12.7 mm) | 1.27% | 0.79% | $3.11 \times 10^9$ mm$^4$ |
| Case-2 | 27.6 | 351 | 483 | 332 | 500 | D10 (9.53 mm) | 1.25% | 0.79% | $3.13 \times 10^9$ mm$^4$ |

$f_c$: Concrete strength on the day of loading test. $f_y{'}$, $f_{hy}{'}$: Average yield strength and average tensile strength of longitudinal reinforcements. $f_u{'}$, $f_{hu}{'}$: Average yield strength and average tensile strength of hoop reinforcements. $D_{lr}$: Diameter of longitudinal reinforcements. $\rho$, $\rho_w$: Longitudinal reinforcement ratio and hoop reinforcement volume ratio. $I_{rc}$: Inertia moment of each section.

## 2.2. Test Setup, Loading Protocol, and Instrumentation

In this study, a cyclic loading test was performed using the loading apparatus depicted in Figure 2. Load and displacement measurements were obtained using load cells (MISUO, DWG.NO. HC-15391, Japan) and LVDTs (displacement meters) integrated into the vertical and horizontal jacks. Additionally, a 160 kN axial force was applied using a vertical hydraulic jack (KYOWA, LUR-B-1MNSA1, Japan).

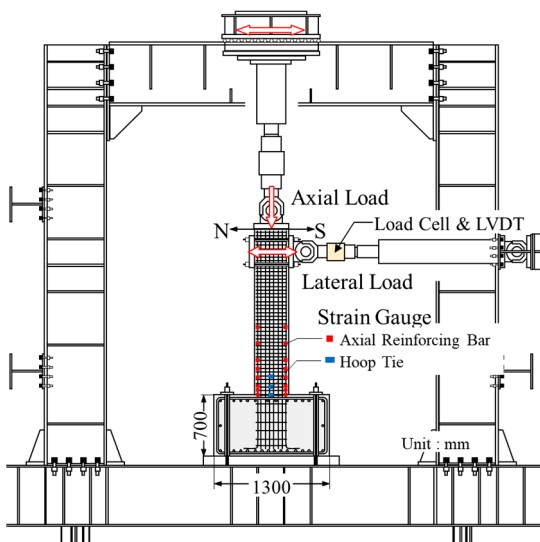

**Figure 2.** Test Setup.

A positive and negative alternating load was applied in the horizontal loading direction by incrementally increasing the displacement. Figure 3 illustrates the horizontal displacements applied during these tests. The reference displacement was set to 0.5% of the effective height for both experimental specimens (hereafter referred to as "0.5% Drift"), resulting in 6.75 mm for Case-1 and 8 mm for Case-2. These values are roughly equal to the design yield displacement, and integer multiples of this displacement were manually applied repeatedly. The number of repetitions was set to three for the same displacement

amplitude. To ensure proper adaptation of the specimen to the loading device, a load was applied once in both positive and negative directions at 0.25% Drift before the loading test commenced.

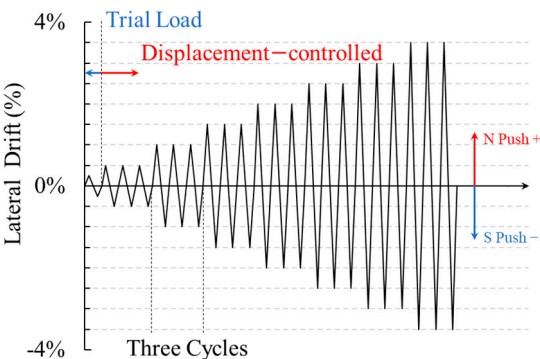

**Figure 3.** Loading Protocol.

Due to experimental constraints, only the damage to the longitudinal reinforcements in Case-2 was examined. Strain gauges were attached to the central longitudinal reinforcements on both the N-face and S-face sides of the loading direction, as illustrated in Figure 4. The strain gauges were placed from the base up to a height of 800 mm, and the strain at each loading step was measured.

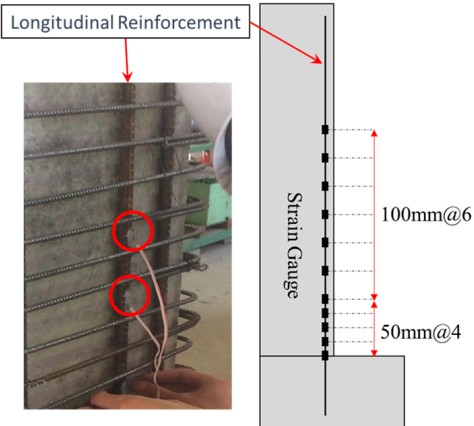

**Figure 4.** Strain Gauge.

### 2.3. Cyclic Behaviors of Specimens

Figure 5 illustrates the load-displacement history of both specimens. In Case-1, the horizontal load-bearing capacity stabilizes after 1% Drift, and the maximum load capacity of 119.8 kN is reached at 3% Drift. Subsequently, it maintains a nearly stable horizontal load-bearing capacity of up to 3.5% Drift. In Case-2, which features a smaller diameter and denser arrangement of longitudinal reinforcements, the maximum horizontal load of approximately 82 kN is reached at 1.5% Drift.

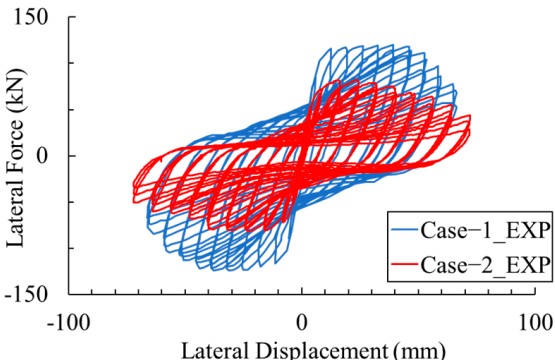

**Figure 5.** Lateral Force versus Lateral Drift Relationships.

As the effective height $h_{ef}$ of both specimens differs, the bending moment at the base of the column (P·$h_{ef}$) is calculated for each specimen, the Moment-Drift history normalized to the maximum bending moment is shown in Figure 6a, and the envelope curves is shown in Figure 6b. The Case-1 specimen maintains a stable load-bearing capacity from 1.0% Drift and withstands up to 3.5% Drift on both positive and negative sides. However, in Case-2, after reaching the maximum horizontal load at 1.5% Drift, the load-bearing capacity begins to degrade, and ultimately only about 40% of the maximum value remains.

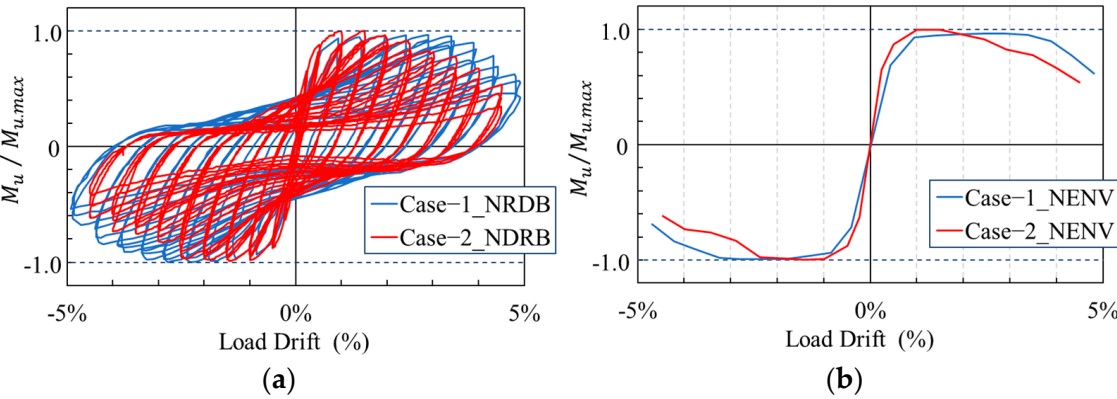

**Figure 6.** Normalized Hysteresis Curves. (**a**) Normalized Bending Moment History. (**b**) Envelope Lines.

Furthermore, the load-displacement history of both specimens exhibits a significant pinching phenomenon. The pinching phenomenon refers to a state in which the stiffness of the RC column decreases during unloading due to the complex material nonlinearity of reinforced concrete when the column is subjected to repeated loading. The degree of pinching of the hysteresis curve is a critical evaluation criterion to evaluate the RC structure for seismic energy absorption, so it is also essential to quantify the pinching phenomenon.

To quantify the pinching phenomenon of both specimens, the stiffness degradation of the specimens was investigated in this research. Figure 7 presents the stiffness degradation history of both specimens at each Drift. To evaluate the normalized stiffness, the stiffness $K_i$ and the yield stiffness $K_e$ at each Drift were employed. $K_i$ is obtained by the ratio of the moment at the column base position to the displacement at the loading position at i% Drift, and $K_e$ is the stiffness when the longitudinal reinforcement yields (1.0% Drift). In both specimens, stiffness decreases as loading progresses, and the stiffness degradation rate of Case-2 with slender reinforcement is faster than that of Case-1. The final stiffness of Case-2 is only half that of Case-1, suggesting that the pinching phenomenon in the load-displacement history of Case-2 is more severe than that of Case-1.

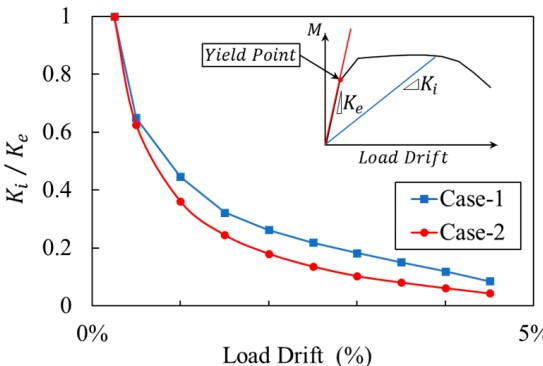

**Figure 7.** Stiffness Degradation History.

Although various causes contribute to the pinching phenomenon in RC columns, in the case of the slender RC columns used in this study, previous research has indicated that the primary cause is a bond failure in the reinforced concrete [1,22,23].

*2.4. Damage to Longitudinal Reinforcements*

In this study, we focused on the reinforcement damage in Case-2 within small-diameter and high-density bar arrangements. Figure 8a shows the strain history of the longitudinal reinforcement on both the north and south sides at the base of the column. Figure 8b,c show the history of horizontal load versus longitudinal reinforcement strain at the same location. From (a) and (b), the south-side longitudinal reinforcement appears to exhibit almost elastic deformation. However, from (a) and (c), the north-side longitudinal reinforcement experiences plasticization up to 20,000 μ after yielding during the first loading at 1.0% Drift and remains in the strain region of approximately +4000 μ (tension) until the loading reaches 2.5% Drift. The extent of plasticization of the longitudinal reinforcement differs drastically between the south and north sides, and this may be due to the bonding failure of the southern steel bars from the beginning, resulting in stress not being well transmitted to the reinforcements, resulting in the reinforcements being in an elastic deformation state. However, since the southern reinforcement is almost unable to bear the load, stress concentration occurs in the northern reinforcement, resulting in significant plastic deformation.

Additionally, as shown in (c), the relationship between the horizontal load and longitudinal strain of the north-side reinforcement shifts to the compression side (negative) from 3.0% Drift onwards, and the horizontal load-bearing capacity gradually decreases as the deformation increases. As the loading progressed, it was observed that the longitudinal reinforcement in the corner began to buckle at 3.5% Drift, as shown in Figure 9a–c. At 4.5% Drift, the sound of the longitudinal reinforcement breaking was confirmed, and a sudden drop in load-bearing capacity was observed, leading to the termination of the experiment.

Similarly to Figure 8, the history of horizontal load versus longitudinal reinforcement strain at 100 mm and 200 mm above the column base is shown in Figure 10 for 0.25% to 3.0% Drift. Focusing on the height of 100 mm above the column base, the damage is less severe than at the column base. However, significant plasticization occurs on the north-side longitudinal reinforcement (with strain exceeding 10,000 μ), and residual strain is also observed.

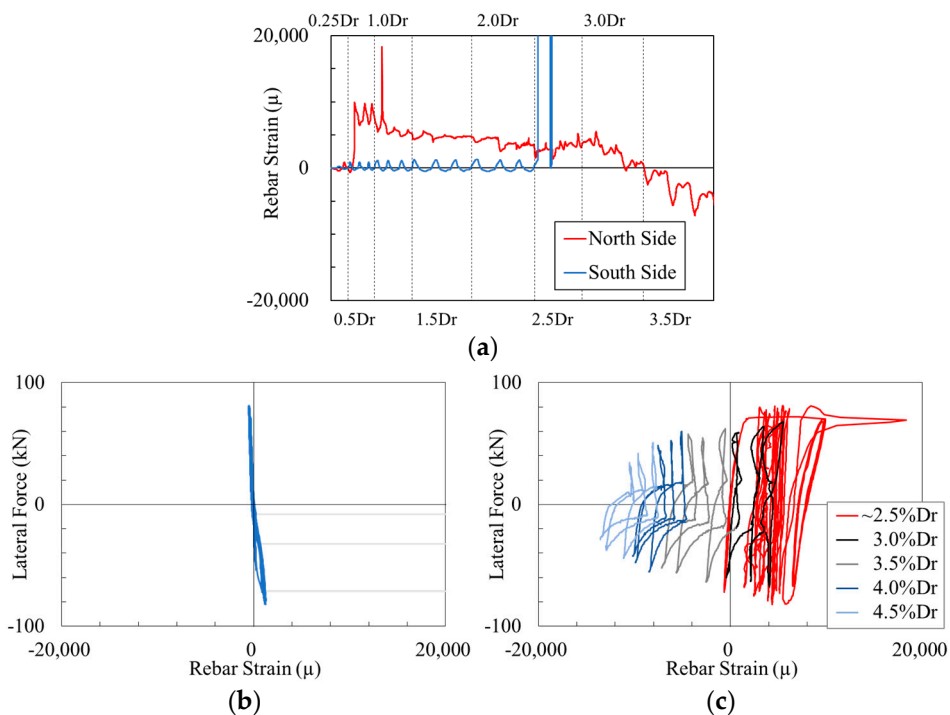

**Figure 8.** Damage to Axial Reinforcements at Base of Columns. (**a**) Strain History. (**b**) Load-Strain History on South Side. (**c**) Load-Strain History on North Side.

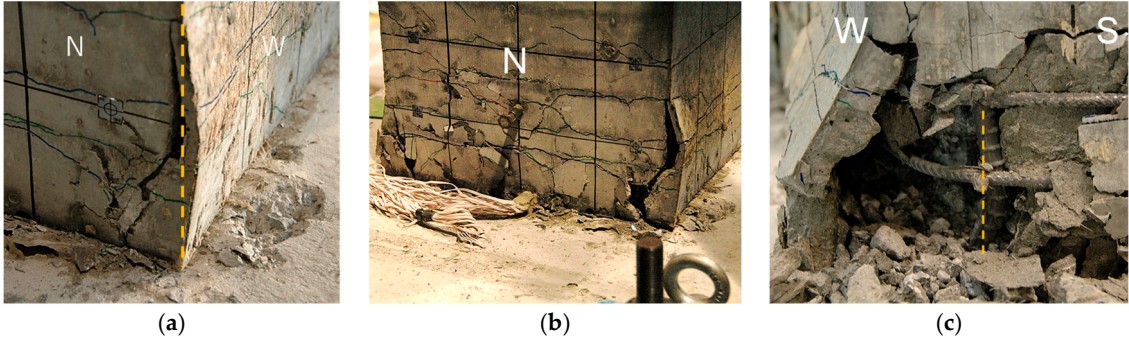

**Figure 9.** Damage at Column Base Location (Case-2). (**a**) 2.0%Drift of NW. (**b**) 2.5%Drift of NW. (**c**) 3.5%Drift of WS.

On the other hand, the plastic deformation of the south-side longitudinal reinforcement is considerably smaller than that of the north-side, as in the case of the column base, and the difference in damage levels between the north and south sides is significant. For the longitudinal reinforcement strain at the height of 200 mm above the column base, the damage level is smaller compared to the column base and 100 mm height position, but the difference in deformation between the north and south sides is almost negligible.

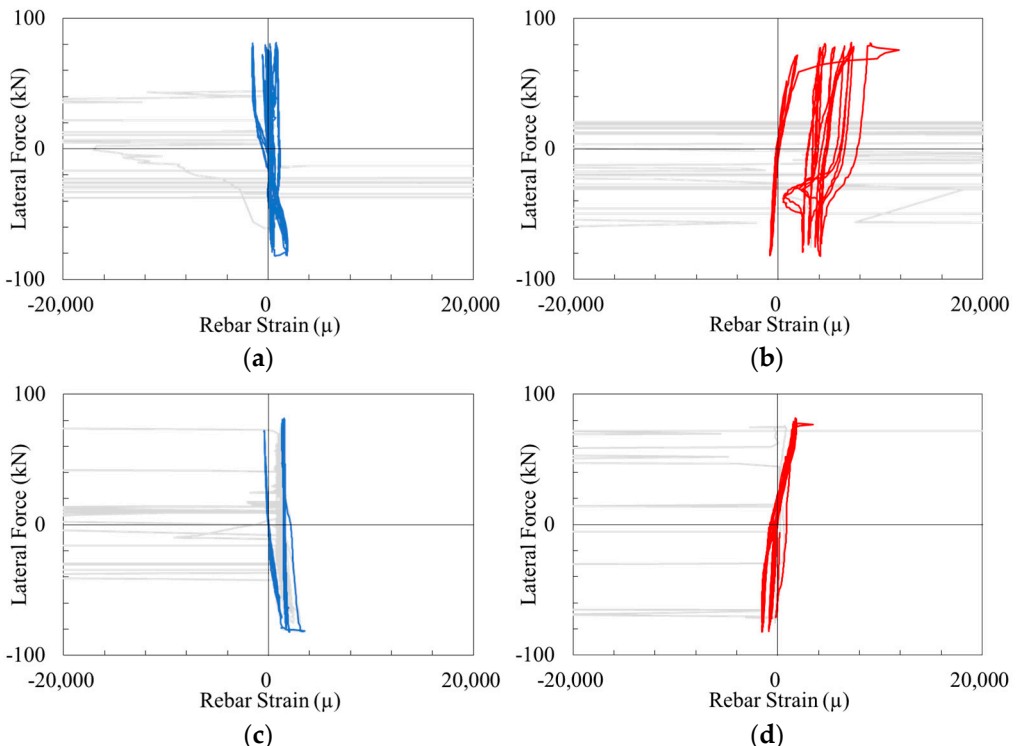

**Figure 10.** Damage to Axial Reinforcements at 100 mm & 200 mm of Columns. (**a**) LS History on South Side (100 mm). (**b**) LS History on North Side (100 mm). (**c**) LS History on South Side (200 mm). (**d**) LS History on South Side (200 mm).

*2.5. Rotational Deformation at the Base of Columns*

As shown in Figure 11, a steel rod was installed at the height of 200 mm from the pier base, penetrating the pier cross-section in the loading direction, and displacement sensors were installed at both ends of the steel rod to measure vertical displacement.

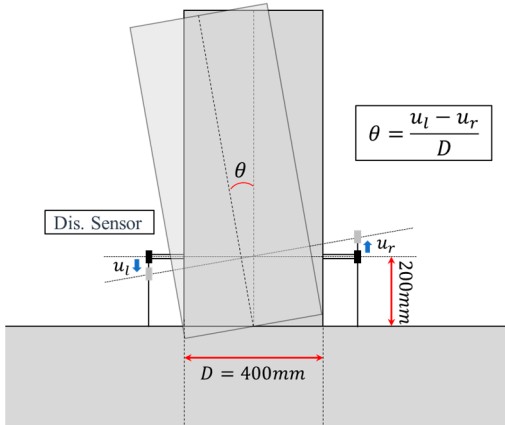

**Figure 11.** Measurement of Rotational Deformation.

Displacement gauges were installed at both ends of a steel rod penetrating the cross-section of the bridge pier to measure the vertical displacement. From these measurements, the rotational angle at the base of the RC column was calculated. Furthermore, based on the calculated rotation angle, the displacement induced by the load position can be determined in relation to the rotational deformation at the column base as follows:

$$u_{fd} = \theta \cdot h_{ef} \qquad (2)$$

Here, $h_{ef}$ represents the effective height of the bridge pier. The rotational angle $\theta$ at the lowest level of the bridge pier is used to approximate the rotational angle induced in the pier body due to the longitudinal reinforcement elongation caused by bond failure inside the footing. Therefore, it is believed that the horizontal displacement $u_{fd}$ in Equation (2) can approximate the horizontal displacement caused by the bridge pier rotation to some extent.

However, it should be noted that the influence of the bending deformation and shear misalignment deformation occurring in the range below the displacement gauge is included in Equation (2). As a result, the actual rotation displacement $u_{fd}$ at the loading point caused by the deformation of the longitudinal reinforcement elongation inside the footing, may be overestimated.

Figure 12 shows the rotational deformation history of both specimens. The horizontal axis represents the loading Drift, and the vertical axis is the rotational angle at the base of the column.

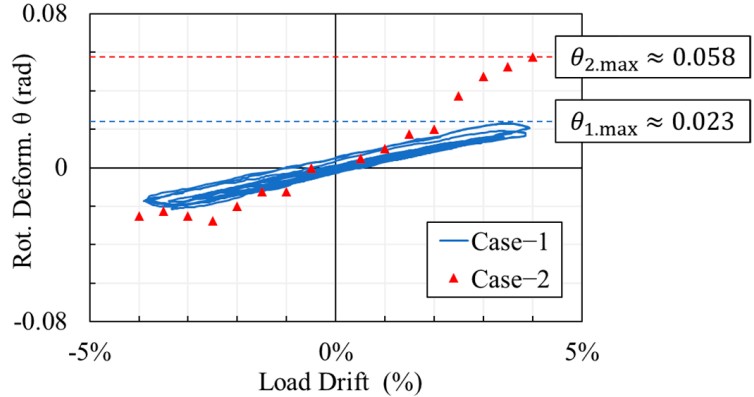

**Figure 12.** Rotational Deformation History.

The rotation deformation history of Case-1 is approximately linear, and when the rotation deformation reaches its maximum value, with approximately 60% of the total deformation $u_{total}$ being caused by the rotational deformation $u_{fd}$. As mentioned earlier since the bending deformation and shear deformation occurring below the displacement meter are included, the actual rotation deformation is smaller than 60%.

In contrast, the red triangular markers representing Case-2 exhibit the rotational deformation at the column base for each Drift level. In the initial stage of the experiment (before 2.0% Drift), Case-2 and Case-1 were same, with the rotational deformation at the base increasing linearly with the load. However, from 2.5% Drift onwards, the positive rotational deformation in Case-2 experienced a drastic increase (approximately doubling), while the negative direction saw a slight decline from −2.5% Drift due to the abrupt increase in the positive rotational deformation. Overall, the decrease was less than 20%. Simultaneously, linear growth was restored in the positive direction from 2.5% Drift to 4.0% Drift. At 4.0% Drift, while the negative rotational deformation in Case-2 remained consistent with Case-1, the positive rotational deformation reached 2.5 times that of Case-1.

Figure 13 illustrates the bond failure mechanism of Case-2 and its subsequent impact. In the initial stage of the load test, the south side reinforcement experienced bond failure due to concrete cracking, preventing stress from being effectively transferred from the concrete to the reinforcement. This is the reason why the south-side reinforcement remained in an elastic state. Simultaneously, because of the bond failure of the south side reinforcement, the stress was concentrated on the north side reinforcement, leading to stress concentration and significant plastic deformation. At 1.0% drift, the north side reinforcement underwent substantial plastic deformation (about 20,000 μ) after the first elongation. Subsequently, cracks at the base of the RC column penetrated the entire column section (as observed in Figure 9, horizontal through-cracks were also observed in the experiment), resulting in bond failure of the north side reinforcement. Consequently, the response to the hysteresis

load was not noticeable (it did not increase or decrease with the load), but instead remained at a certain residual deformation (about 4000 μ).

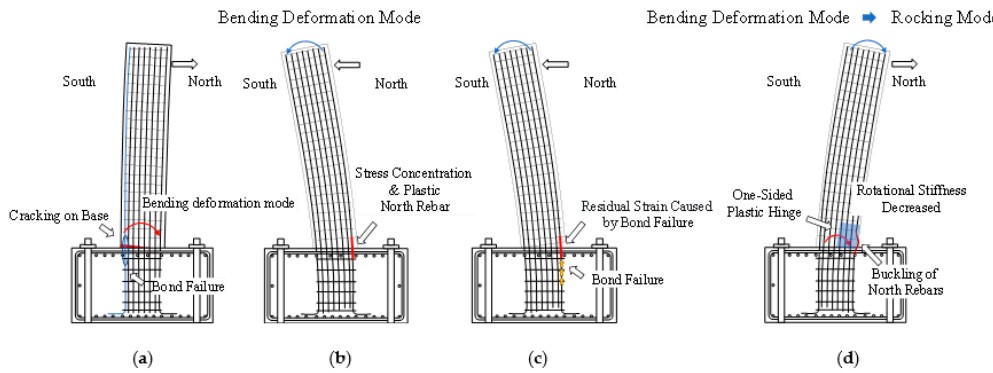

**Figure 13.** Mechanism of Bond Failure (Case-2). (**a**) 0.5% Drift. (**b**) 0.5~1.0% Drift. (**c**) 0.5~1.0% Drift. (**d**) After 2.5% Drift.

After 2.5% drift, as the deformation of the RC column further increased, the north side reinforcement, which was pulled out from the footing, first experienced buckling due to compressive stress (Figure 9b). Therefore, the strain measured by the reinforcement strain gauge gradually shifted to the negative direction (compression), leading to the formation of a unidirectional plastic hinge at the base of the RC column. The rotational stiffness dropped sharply, which leads the RC column to a change in the failure mode from the bending deformation mode to the rocking deformation mode. As shown in Figure 12, after 2.5% drift, the rotational deformation in the positive direction increased sharply, while the rotational deformation in the opposite direction did not change significantly.

From the above, it can be concluded that, compared to Case-1, Case-2 exhibited a difference in the bond failure situation at the column base during loading, even with a similar amount of longitudinal reinforcement, and the load-bearing deformation capacity of the column decreased. Additionally, by using small-diameter with high-density bar arrangements, the rocking phenomenon became more prominent in Case-2 due to the buckling and splitting bond failure of the reinforcement.

## 3. FEA Results without Bond-Slip

In this study, we utilize general-purpose structural analysis software DIANA to conduct a finite element analysis considering only material nonlinearity and disregarding bond failure of the reinforcement.

### 3.1. Structure Modeling

The structural model is shown in Figure 14. In this study, we focus on material nonlinearity and reproduce the hysteresis characteristics of RC columns subjected to unidirectional loading. We disregard shear and torsional deformations in the depth direction. To reduce the computation cost, we use 4-node isoperimetric plane stress elements Q8MEN based on linear interpolation and Gauss integration [24].

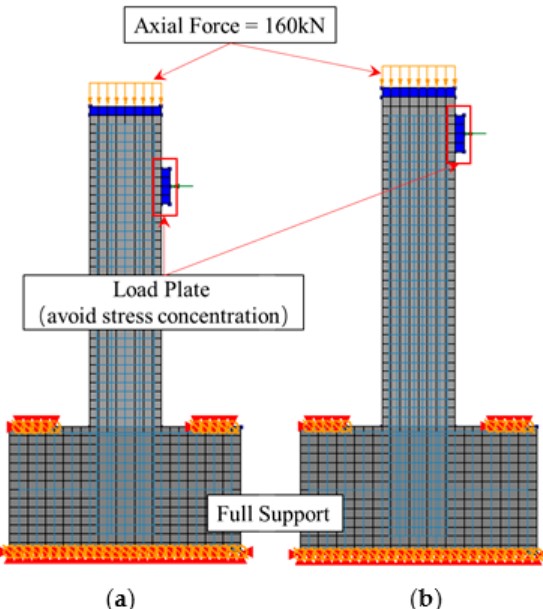

**Figure 14.** FE Model. (**a**) Case-1. (**b**) Case-2.

In the discrete reinforcement model, it is necessary to connect the ends of elements to each other. In typical RC structures with complex reinforcement arrangements to significantly increase the number of elements, the surrounding concrete elements need to be sufficiently small, however, making the concrete elements too small may lead to stress concentration-induced failures [25].

To investigate the bond-slip phenomenon, it is possible to reproduce the bond-slip phenomenon by introducing interface elements between the reinforcement and concrete, which will be described in detail in the later. Therefore, this study uses embedded reinforcement elements (Bar Type) as shown in Figure 15, the elements and reinforcement can be defined independently from each other and refer to unique geometric definitions. Typically, according to the assumption of perfect adhesion between reinforcement and concrete, the strain of embedded reinforcement elements is calculated from the displacement field of the host elements [26].

reinforced concrete element

$$[K_s] = \int_{L_j} [B]^T \{T\}_j{}^T \{T\}_j [B] A_j E_{sj} dL$$

$$\{T\}_j = \{\cos^2 \phi_j \ \sin^2 \phi_j \ \cos \phi_j \ \sin \phi_j\}$$

$$[K_e] = [K_c] + \sum_{j=1}^{m} [K_s]_j$$

**Figure 15.** Embedded Reinforcement Elements.

### 3.2. Material Modeling of Concrete

First, the material nonlinearity of the concrete model with plane stress elements is required to evaluate the reduction of tensile strength and shear strength associated with crack formation, localization of damage, and the history of unloading and reloading.

Maekawa et al. [27,28] proposed a mathematical model based on substantial experimental data from cyclic loading tests on reinforced concrete, addressing key characteristics of concrete behavior under cyclic loading. This model considers not only the opening and closing of cracks during the loading process and the shear properties at the crack locations but also the lateral expansion effect caused by the Poisson's ratio and the influence of lateral restraint from the steel reinforcement, most importantly, the model can better reflect the objective energy dissipation during the unloading and reloading process. Consequently, it has been extensively applied in finite element analysis of RC structures subjected to hysteretic loading.

This study adopts a multi-directional fixed crack model implemented in DIANA. This model is a distributed crack-type nonlinear model based on the Maekawa-Fukuura Model, the model extends the total strain crack model by incorporating an elastic-plastic failure model and the active crack method. The analysis starts with the elastic-plastic failure model, once the principal tensile stress reaches the tensile strength or the unloading stress, the analysis switches from the original elastic-plastic failure model to the total strain crack model and cannot return to the elastic-plastic failure model. New cracks at each integration point can be determined according to the flow shown in Figure 16.

After crack formation, the uniaxial stress-strain relationship of the concrete changes according to the Maekawa failure curve shown in Figure 17 [28], and the JSCE softening model is adopted in the tensile zone. Regarding the compressive degradation characteristics of cracked concrete, the model defined by the JSCE 2012 (Re) model [29] is used to reduce the compressive strength of concrete material caused by transverse cracks in the principal compressive stress direction. As shown in Figure 18, the horizontal axis represents the maximum tensile total strain in the loading cycle, and the vertical axis represents the corresponding reduction factor of compressive strength.

### 3.3. Material Modeling of Reinforcements

In this study, the constitutive laws for longitudinal reinforcement were adopted, considering both the curvilinear model which faithfully accounts for the Bauschinger effect during positive and negative cyclic loading, and the bilinear model, known as the Menegotto-Pinto Model [30]. As shown in Figure 19, the model uses the stress $\sigma_r{}^n$ and strain $\varepsilon_r{}^n$ at the load reversal point of the final load history to non-dimensionalize the stress and strain as $\varepsilon^*$ and $\sigma^*$, respectively.

The input parameters of the model include initial elastic modulus, yield stress, strain hardening ratio, the initial curvature parameter ($R^0$), and two empirical parameters ($A_1$ and $A_2$) representing the cyclic stiffness degradation of the stress-strain behavior (Bauschinger's effect). According to data given in another paper [31], The parameters describing the cyclic stiffness degradation characteristics of the reinforcements are calibrated as $R^0 = 20.0$, $A_1 = 18.5$, and $A_2 = 0.15$.

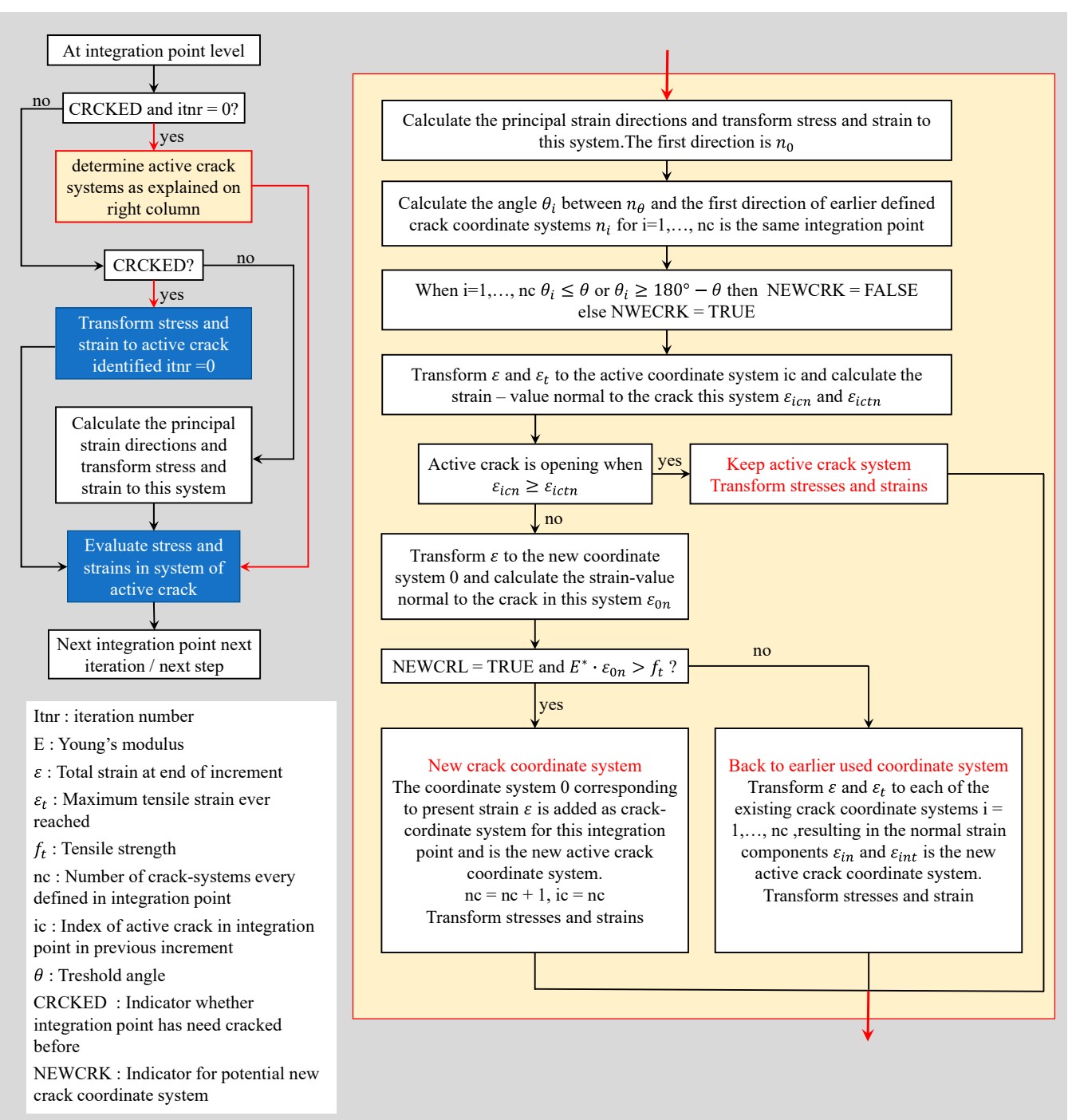

**Figure 16.** Crack System for FE analysis.

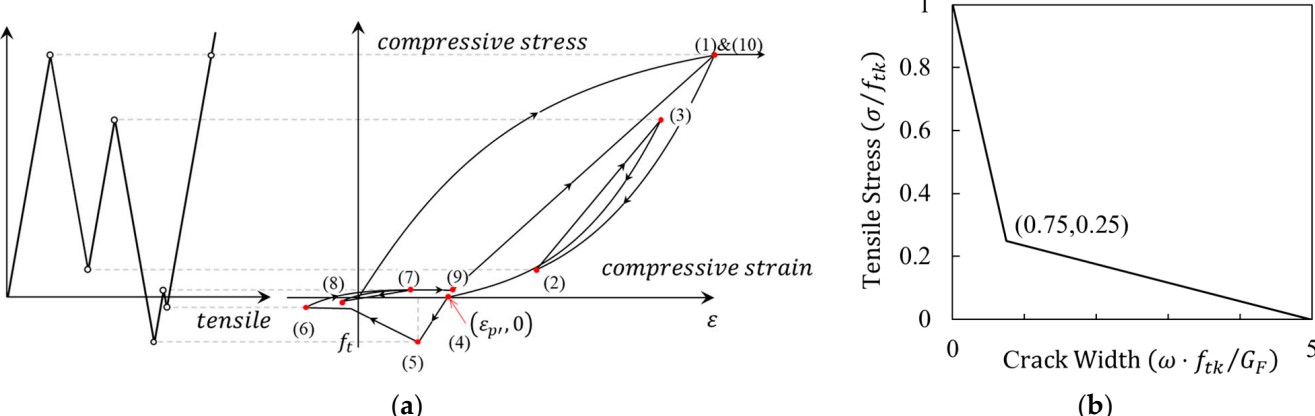

**Figure 17.** Hysteresis for Maekawa model. (**a**) Maekawa Crack Curve. (**b**) JSCE Softening Model.

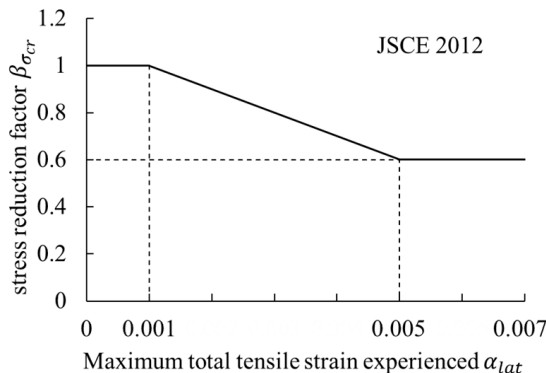

**Figure 18.** Stress Reduction Factor due to Lateral Cracking.

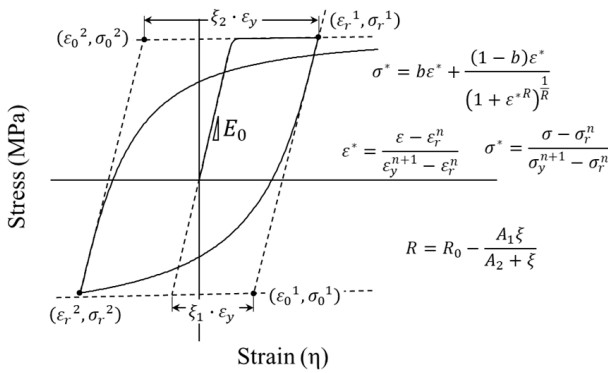

**Figure 19.** Menegotto-Pinto Model.

### 3.4. Discussion of Analysis Results without Bond-Slip

In summary, the horizontal load-displacement relationship and envelope at the load point were obtained from the analysis of the model that assumes fully bonded between the longitudinal reinforcement and concrete are shown in Figure 20. Since the geometrical nonlinearity of the reinforcement was not considered, the analysis results of the basic model Case-1 could not reproduce in load-bearing capacity drop due to the spalling of the covering concrete and buckling of the longitudinal reinforcement of the Case-1 specimen from 4.0% Drift, however, it successfully reproduced the maximum and minimum horizontal forces up to 3.5% Drift before the reinforcement buckled. And for Case-2, as in Case-1, although the maximum and minimum horizontal forces were reproduced up to 2.0% Drift on the

front side and 2.5% Drift on the reverse side, the subsequent drop in load capacity was not reproduced.

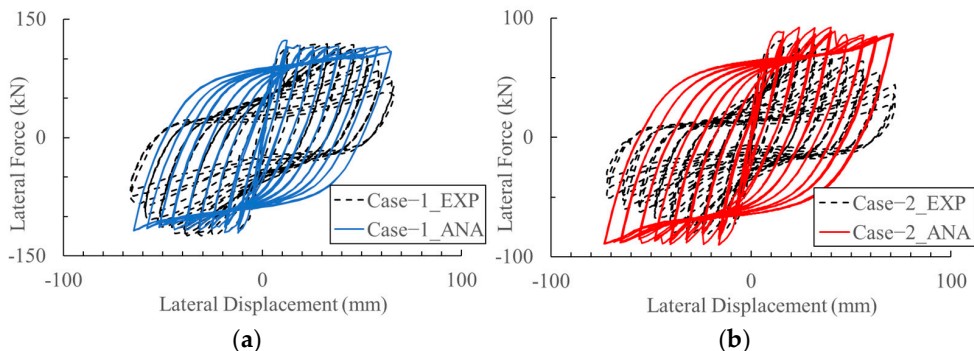

(**a**)
(**b**)

**Figure 20.** Examples of Force–Displacement Responses without Bond-Slip. (**a**) Case-1(EXP vs. ANA). (**b**) Case-2 (EXP vs. ANA).

For both specimens, the second and fourth-quadrant load-displacement histories of the horizontal load-displacement relationship were significantly inflated compared to the experimental results. The result showed an overestimation of the energy absorption capacity of the RC column since the reverse S-shaped curve observed due to stiffness changes during unloading and reloading was not reproduced.

At the first cycle of 3.5% Drift when the maximum positive displacement is reached, the crack contour diagrams of the analysis are shown in Figure 21. For both specimens, the damage is concentrated from the boundary to a height of 200 mm, which is generally consistent with the damaged area during the experiment. The analysis results of Case-1, horizontal cracks appear at the tension edge and extend to the compression edge, but the horizontal cracks have not penetrated yet. The cracks at the compressive edge are mainly distributed in the longitudinal reinforcement direction. On the other hand, in the analysis results of Case-2 with closely spaced reinforcement, the horizontal cracks at the boundary have penetrated wholly, and the damage level is higher than in Case-1.

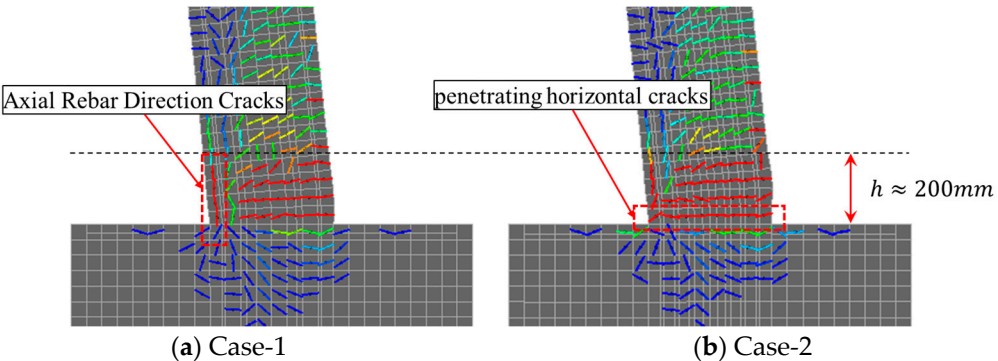

(**a**) Case-1
(**b**) Case-2

**Figure 21.** Crack contour diagrams(0–200 mm, 3.5%Drift).

Moreover, the analysis results of the Case-2 specimen with small-diameter and high-density arrangements, and the horizontal load-longitudinal reinforcement strain history for are shown in Figure 22. In the experimental results shown in Figure 8 (strain history at the 0 mm position), the south-side reinforcement exhibited almost elastic deformation. In contrast, in this analysis case, both the north and south-side longitudinal reinforcements have yielded. Although the load-displacement relationship is generally consistent, it can be concluded that the damage condition of the longitudinal reinforcement has not been reproduced accurately.

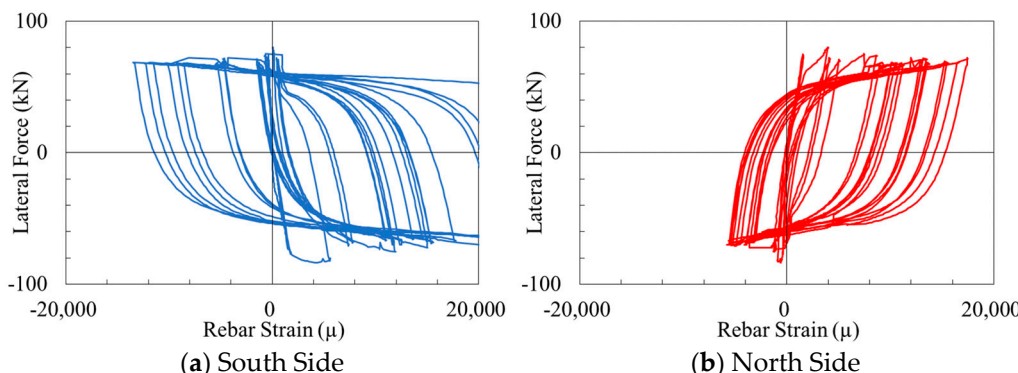

**Figure 22.** Damage to Axial Reinforcements at Base of Columns by Analysis without Bond-slip (Case-2).

In summary, the study aims to analyze the behavior of reinforced concrete columns under various loading conditions by using the model with the assumption of the complete fixation between longitudinal reinforcement and concrete. Some limitations were observed while the results successfully reproduced the maximum and minimum horizontal forces up to a certain level of Drift for both Case-1 and Case-2 specimens. Specifically, the analysis could not accurately reproduce the decrease in load-bearing capacity of the Case-1 specimen due to the spalling of the covering concrete and buckling of the longitudinal reinforcement, and the load-bearing capacity drop of the Case-2 specimen due to bond failure.

Additionally, the analysis overestimated the energy absorption capacity of the RC columns, as it could not reproduce the reverse S-shaped curve observed in the experiment induced by bond-slip. Furthermore, although the horizontal load-displacement relationship and the damage patterns were generally consistent with the experimental results, the damage condition of the longitudinal reinforcement in Case-2 was not accurately reproduced. To improve the accuracy of the analysis, these findings highlight the importance of considering the bond-slip phenomenon and geometric nonlinearity of the reinforcement in future research.

## 4. Effects of Different Mode Bond-Slip Fracture

In this chapter, the finite element models of the experimental RC column were developed using the finite element analysis software DIANA, and an appropriate bond-slip model was employed to verify the numerical model based on experimental results. The impact of different bond failure modes on the overall deformation load capacity of the RC column and internal damage was subsequently investigated.

### 4.1. Research Review on Modeling of Bond-Slip Phenomena

In the analysis of RC beam shear resistance conducted by D. Ngo and A.C. Scordelis in 1967 [32], reinforcements and concrete were modeled separately. Crack propagation was considered using linkage elements, and the bonding between the reinforcements and concrete was taken into consideration through modeling.

Subsequently, to enhance the simulation capabilities of link elements, researchers proposed a spring-based modeling method that employs multiple spring elements to connect two material surfaces to simulate the bond-slip behavior between materials [33].

With the advancement of finite element techniques, researchers introduced more sophisticated modeling methods, such as interface elements. These specialized finite elements simulate the bond-slip behavior between material surfaces by utilizing nonlinear and multidirectional intrinsic structural relationships. They provide a more accurate representation of complex bond and slip phenomena and allowed automatic calculation of material parameters.

The numerical models for bond stress-slip between reinforcement and concrete have been investigated in various ways, from the earliest single linear models to later bilinear and nonlinear models. Finally, the numerical model that accounts for a reduction in bond strength due to bond damage is the most prevalently employed approach in the field.

Eligehausen proposed a bond-slip model for cyclic loading based on extensive experiments, which is still widely used today [4]. The numerical bond-slip model employed in this study is founded on Eligehausen's model, which modifies the CEB-FIP 1990 Model of the European International Concrete Committee. This modification adopts the CEB-FIP 2010 Model [34] and incorporates a softening equation for bond strength due to reinforcement strain and an increasing equation for bond strength due to lateral restraint.

### 4.2. Numerical Model of Bond-Slip

In reinforced concrete, the interaction between the reinforcement and the concrete is highly complex. The interaction is governed by secondary transverse and longitudinal cracks in the vicinity of the reinforcement. This behavior can be modeled with a bond-slip mechanism where the relative slip of the reinforcement and the concrete is described in a phenomenological sense. The mechanical behavior of the slip zone is then described by the interface element with a zero thickness, as shown in Figure 23. The constitutive laws for bond-slip which have been proposed are mostly based on the total deformation theory, which expresses the tractions as the function of the total relative displacements. In Diana, the relationship between the normal traction and the normal relative displacement is assumed to be linearly elastic, whereas the relationship between the shear traction (bond stress) and the slip is assumed to be a nonlinear function.

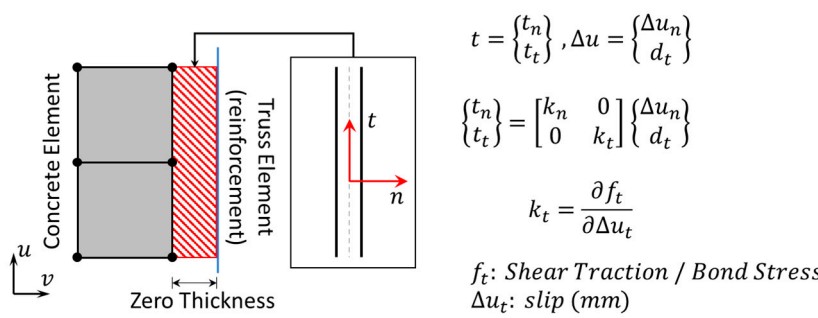

$$t = \begin{Bmatrix} t_n \\ t_t \end{Bmatrix}, \Delta u = \begin{Bmatrix} \Delta u_n \\ d_t \end{Bmatrix}$$

$$\begin{Bmatrix} t_n \\ t_t \end{Bmatrix} = \begin{bmatrix} k_n & 0 \\ 0 & k_t \end{bmatrix} \begin{Bmatrix} \Delta u_n \\ d_t \end{Bmatrix}$$

$$k_t = \frac{\partial f_t}{\partial \Delta u_t}$$

$f_t$: *Shear Traction / Bond Stress*
$\Delta u_t$: *slip (mm)*

**Figure 23.** Bond-Slip Interface Element.

In this study, the nonlinear function between bond stress and slip is represented by the CEB-FIP 2010 model described earlier, since the model adequately considers the variation of bond stress under cyclic loading. The Bond-slip unloading/reloading of FIB Model Code 2010 is built into Diana, as shown in Figure 24a.

In the first part, the power function of the bond-slip stress $\tau$ is developed from 0 to the maximum bond-slip stress $\tau_{max}$ at the relative slip displacement $s_1$. In the second part, the bond-slip stress $\tau$ remained constant at $\tau_{max}$ until the relative slip displacement $s_2$. In the third part, the bond-slip stress $\tau$ is decreased linearly to the ultimate bond-slip stress $\tau_f$ at the relative slip displacement $s_3$. After the relative slip displacement $s_3$ the bond-slip stress $\tau$ remained constant at $\tau_f$.

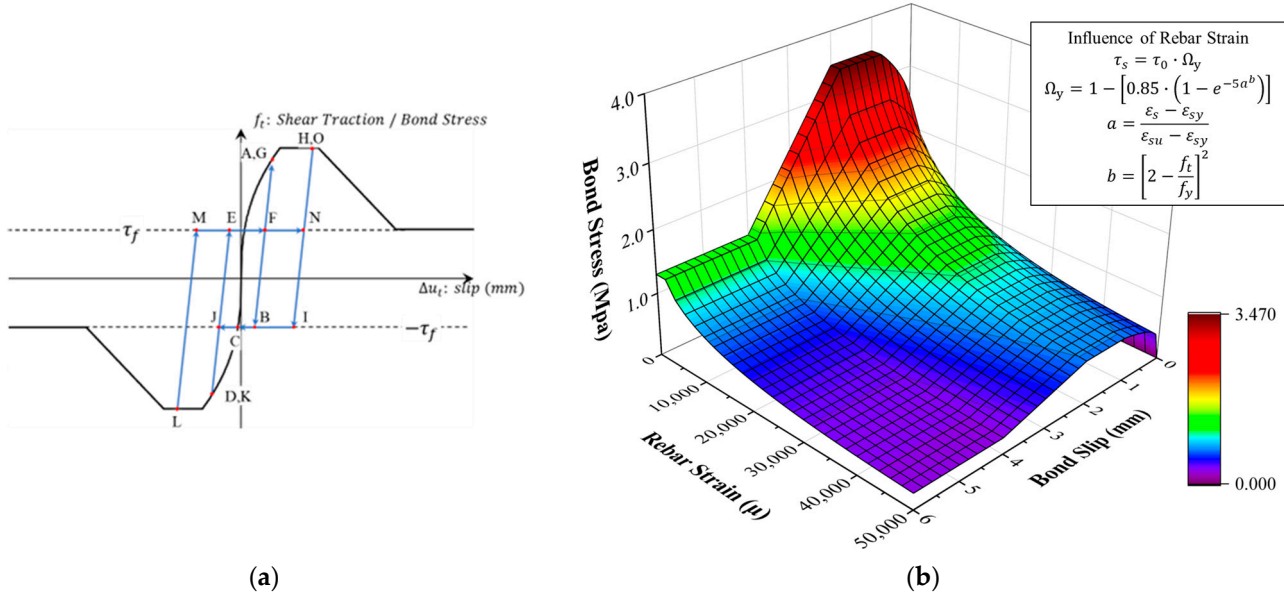

**Figure 24.** CEB-FIP 2010 Bond-Slip Model. (**a**) Bond-slip unloading/reloading Model. (**b**) Bond Stress-Slip-Strain 3D Model.

Furthermore, the original CEB-FIP 2010 model considers the effect of plastic deformation of reinforcement on bond strength, forming a 3-dimensional bond stress-slip-strain model, as illustrated in Figure 24b. As the reinforcement deforms plastically, the bond strength decreases exponentially. When the reinforcement strain reaches 100,000, the bond strength is approximately only 10% of the original. However, the built-in model in Diana only considers cyclic loading and does not account for the effects of reinforcement plasticity. Therefore, based on the result of reinforcement damage measured in the Case-2 experiment, considering the mid-term elongation of approximately 4000 in the north side reinforcement, we established a new bond-slip cyclic model with a maximum bond-slip strength which is around 3.47 MPa.

### 4.3. Investigation of Different Bond-Slip Failure Modes Based on FEA

4.3.1. Pull-Out Failure Mode

Pull-out failure is one of the most common forms of bond failure in reinforced concrete (RC) structures, primarily caused by the reduction in bond strength due to concrete damage. When the reinforcements sustain tension, spiral-shaped micro-cracks will form around the ribbed bars, eventually leading to full penetration along the longitudinal direction of the reinforcements and being pulled out [35].

In practical RC cyclic loading tests, it is challenging to determine the root cause of foundation cracks. To address this issue, the research first employs a pull-out bond-slip model for analysis, with the numerical model illustrated in Figure 25 and parameters of model is refer to Table 2. According to the CEB-FIP guidelines, the research obtains the maximum bond stress $\tau_{max} = 3.47$ MPa based on the average concrete strength of 27.6 MPa obtained from the compression tests of the concrete material used and considering the average strain at the yield of the reinforcements and the influence of stirrups. After the bond failure of the reinforced concrete, the residual bond strength $\tau_f = 0.4 \cdot \tau_{max} = 1.388$ MPa.

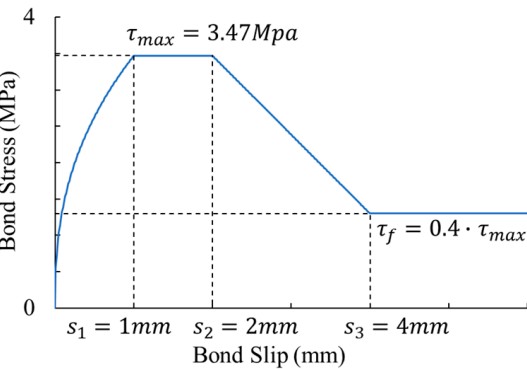

**Figure 25.** CEB-FIP pull-out model.

**Table 2.** Parameters of Pull-Out Model.

| $s_1$ | $s_2$ | $s_3$ | $\tau_{max}$ | $\tau_f$ |
|---|---|---|---|---|
| 1 mm | 2 mm | 4 mm | 3.47 Mpa | 1.38 Mpa |

$s_1$: Slip when reaching maximum bond strength. $s_2$: Slip at the start of bond strength drop. $s_3$: Slip when reaching minimum bond strength.

Upon applying the aforementioned bond model, the analytical results for both specimens are shown in Figure 26. For the Case-1 specimen, although the maximum and minimum load capacities are slightly different compared with the experimental results, the stable load capacity is maintained up to 3.5% Drift, which is the same as the experiment. Furthermore, the load capacity degradation starting at 4.0% Drift is also replicated. During the loading and unloading processes, the changes in stiffness are well represented compared to the hysteresis curve which does not consider the bond-slip shown in Figure 20a. The pinching phenomenon in the hysteresis curve of the RC column is also accurately reproduced, enabling a more precise evaluation of the energy absorption capacity of the RC column during an earthquake.

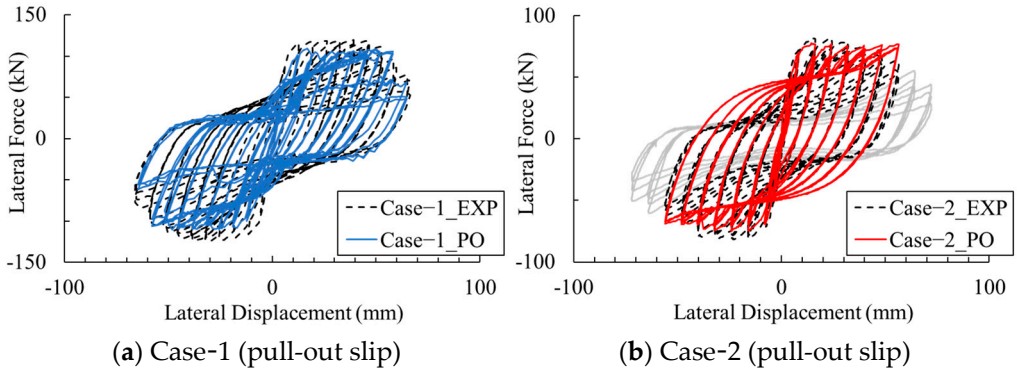

(**a**) Case-1 (pull-out slip)        (**b**) Case-2 (pull-out slip)

**Figure 26.** Examples of Force–Displacement Responses for Pull-Out slip model.

As shown in Figure 26b, the analytical results of the Case-2 specimen are generally similar to those of the previous chapter, where the reinforcement and concrete are assumed to be perfect bonding. However, the reproducibility of the maximum and minimum load capacity has increased compared to the results in the previous chapter. Although the pinching phenomenon of the hysteresis curve is not well reproduced, the stiffness of the RC column during the initial unloading and reloading is essentially consistent with the experimental results.

On the other hand, when using the pull-out bond-slip model, as shown in Figure 27, the strain history of the reinforcement on both sides of the bottom of the Case-2 specimen.

The analysis results of the reinforcement strain on both sides are much smaller in both the tension and compression domains compared to Figure 22, which is consistent with the experimental results shown in Figure 8c. However, the strains of the north and south reinforcements are basically symmetric, and the asymmetry of damage to the reinforcements on both sides, as observed in the experiment, is not reproduced.

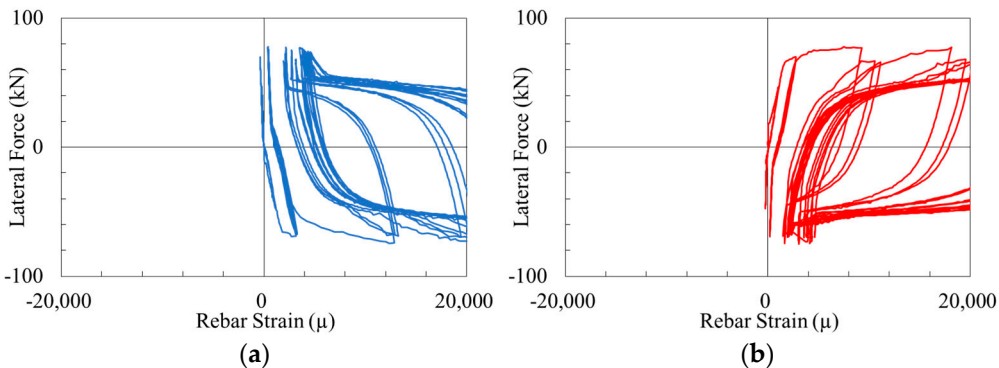

**Figure 27.** Damage to Axial Reinforcements at Base of Columns by Analysis for Pull-Out slip model (Case-2). (**a**) South Side. (**b**) North Side.

The analysis overestimates the influence of the contact area between the reinforcement and the concrete on the bond strength and neglects the insufficient bonding between the small diameter reinforcement ribs and the concrete. This result leads to a situation resembling a perfect bond, which in turn overestimates the overall load-bearing performance and seismic resistance of the RC column.

### 4.3.2. Splitting Failure Mode

In the experiments, the diameter and arrangement density of the longitudinal reinforcement in RC columns is closely related to the bond performance between the reinforcement and concrete. The high diameter of the longitudinal reinforcement, the smaller the frictional force per unit length on the surface of the reinforcements, and is more prone to pull-out failure. When the diameter of the longitudinal reinforcement is smaller, the frictional force per unit length on the surface of the reinforcements is more significant, resulting in higher bond strength and a higher likelihood of splitting failure [36]. Furthermore, when the arrangement of the longitudinal reinforcement is too dense, the stress concentration within the concrete structure is more pronounced, which also contributes to the higher probability of splitting failure [37].

Therefore, we assume that the failure mode of the Case-2 specimen, with densely arranged small-diameter reinforcement, is splitting failure (splitting failure typically results in transverse cracks through the entire concrete structure, which is consistent with the experimental observations in Figure 9).

Figure 28 shows the splitting bond failure model used in this analysis with the same maximum bond strength as in the previous section. In the CEB model, the most significant difference with splitting is the absence of a maximum bond stress maintenance interval (S1–S2) compared to the pull-out model. Upon reaching the maximum bond strength, plastic failure occurs immediately, and the bond strength drops directly to the minimum value.

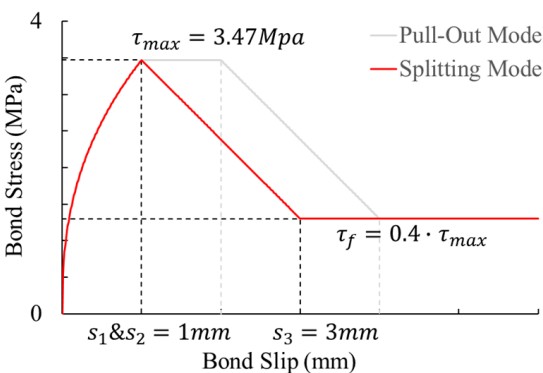

**Figure 28.** CEB-FIP Splitting model.

The hysteresis curve obtained from the analysis of Case-2 using the splitting bond model is shown in Figure 29 and parameters of model is refer to Table 3. Although the horizontal load capacity at each Drift is smaller than the experimental results after 1.0% Drift, the maximum load capacity is accurately reproduced. On the positive side, the degradation trend of load-bearing performance after reaching the maximum horizontal force is also replicated. The pinching behavior during unloading and reloading is not sufficiently reproduced. However, generally capturing the history of the bending deformation behavior observed in the experiments shows the replication of the stiffness degradation during unloading and reloading.

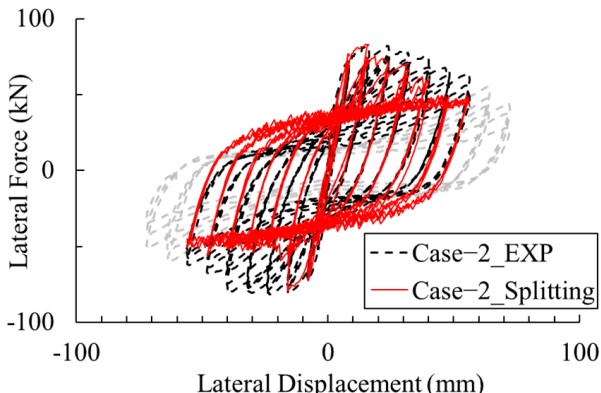

**Figure 29.** Examples of Force–Displacement Responses for Pull-Out slip model.

**Table 3.** Parameters of Splitting Model.

| $s_1$ | $s_2$ | $s_3$ | $\tau_{max}$ | $\tau_f$ |
|---|---|---|---|---|
| 1 mm | 1 mm | 3 mm | 3.47 Mpa | 1.38 Mpa |

$s_1$: Slip when reaching maximum bond strength. $s_2$: Slip at the start of bond strength drop. $s_3$: Slip when reaching minimum bond strength.

Moreover, the history of horizontal load and longitudinal reinforcement strain are shown in Figure 30. While the north-side longitudinal reinforcement exhibits elastic deformation, the south-side longitudinal reinforcement shows permanent strain on the tensile side after plasticization. In the experiment, as shown in Figure 8, the damage was concentrated on the north-side longitudinal reinforcement rather than the south side. Apart from the reversed damage situations between the north and south sides, the experimental results are accurately reproduced.

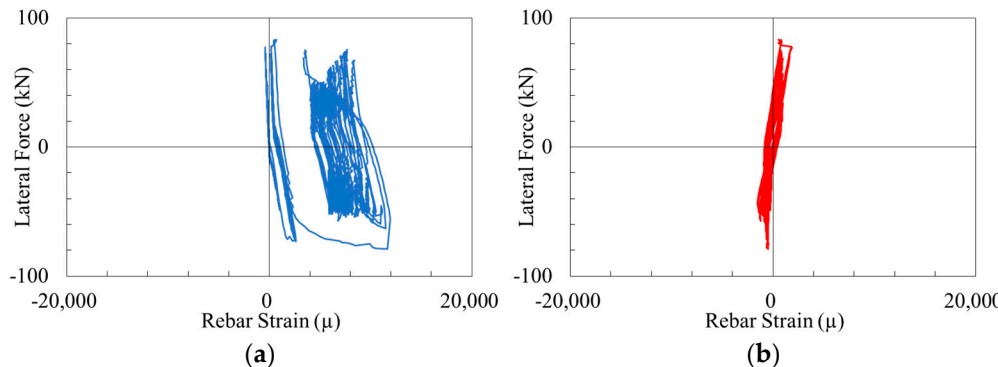

**Figure 30.** Damage to Axial Reinforcements at Base of Columns by Analysis for Splitting slip model (Case-2). (**a**) South Side. (**b**) North Side.

Regarding the rotational deformation of the Case-2 RC column, the comparison between the experimental and analytical results is shown in Figure 31. Although the analytical results do not exhibit a sudden drastic rotational deformation at a certain moment, the maximum value in the positive direction reaches 3.1%, while the negative direction only reaches −2.4%, as shown in Figure 31b. Similar to the experiment, the tendency to shift in one direction is well reproduced. In the analysis, the rotational deformation angle of the base reaches 0.03 at a 3.5% drift, with the corresponding rotational deformation accounting for approximately 85% of the total deformation. Consistent with the experiment, the RC column featuring small-diameter, high-density bar arrangements transition from the bending deformation mode to the rocking deformation mode during the hysteretic loading process.

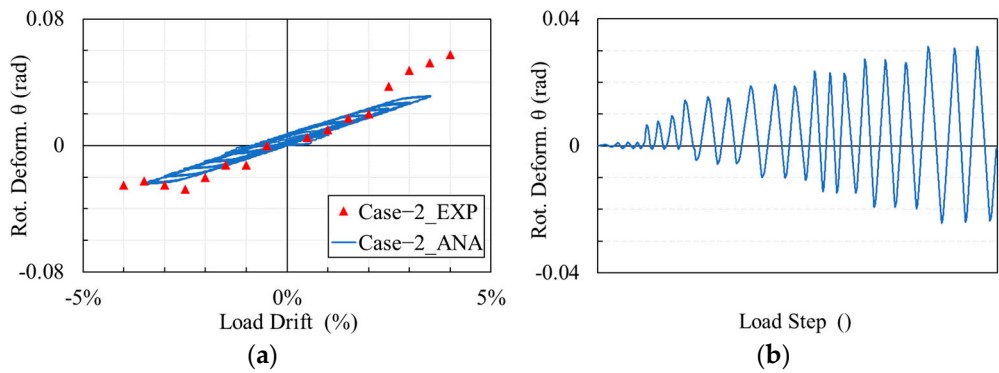

**Figure 31.** Reproducibility of rotational deformation (Case-2). (**a**) Rotational Deformation Drift History. (**b**) Rotational Deformation Step History.

This is because a 2D model is used in this study, and all the reinforcements on one side were simulated with only one bar in the FE model. Therefore, when the bond failure occurs, resulting in a decrease in load-carrying capacity, bond failure occurs simultaneously in all reinforcements on one side However, only some of the reinforcing bars may undergo splitting failure and be pulled out of the footing, while the remaining bars still retain some load-carrying capacity. This is also the reason why in the simulation of the hysteresis curve the load capacity of the RC column on one side does not gradually decrease but drops suddenly after reaching the maximum load capacity. Further studies based on FE simulation analysis using 3D models are required to verify partial bond-slip failure on RC column performance.

Table 4 presents a comparison of the maximum and minimum lateral force from the experimental and analytical results for Case-1, which employs the pull-out model, and Case-2, which employs the splitting model. For Case-1, the error range for the maximum and minimum horizontal loads is approximately 8% to 11% after implementing the pull-out

model. Meanwhile, for Case-2, which adopts the splitting model, the error rate for the maximum and minimum horizontal loads is maintained below 3%. As previously mentioned, although the pinching phenomenon in Case-1 is challenging to quantify, the analytical results reproduce the characteristics of the experimental results well when comparing the hysteresis curve contours. For Case-2, while the accuracy of the hysteresis curve reproduction can be further improved, the pinching phenomenon in the hysteresis curve, the asymmetric rotational deformation history, and the significantly different reinforcement strain histories in the north and south have been well reproduced to some extent. Therefore, we believe that the two bond-slip models proposed in this study, based on the form of bond failure, are both correct and effective.

**Table 4.** Comparison of Experimental and Analytical Results for Lateral Force.

|   | Case-1 (PULL-OUT) | | | Case-2 (SPLITTING) | | |
|---|---|---|---|---|---|---|
|   | EXP | ANA | ERRO | EXP | ANA | ERRO |
| + | 119.81 kN | 109.30 kN | 8.77% | 82.20 kN | 83.09 kN | 1.08% |
| − | −124.21 kN | −110.36 kN | 11.16% | −81.43 kN | −78.92 kN | 3.08% |

## 5. Conclusions

In this study, based on the cyclic loading tests and the finite element analysis, the load-bearing capacities and deformation performances were carefully analyzed for two different reinforced concrete columns with different diameters and reinforcement bar arrangements but with the same longitudinal reinforcement ratios. And three types of bond-slip models—full bonding, pull-out bond failure, and splitting bond failure-were used to analyze the effects on the mechanism and deformation behavior of RC column bending damage. The obtained conclusions from this study are as follows:

- The load-bearing capacity and deformation performance of reinforced concrete columns is significantly affected by the diameter and arrangement of reinforcing bars, despite the longitudinal reinforcement ratios are same. The reinforced concrete column showed a 3.5% drift of high deformation capacity when using large-diameter and low-density bar arrangements. However, significant degradation of the deformation capacity to a 1.5% drift was observed in the reinforced concrete column with the small-diameter and high-density bar arrangements.

- In the cyclic loading test, despite being subjected to symmetrical hysteretic loadings, the RC column with small-diameter and high-density bar arrangements results in the splitting bond failure of the reinforcements. And in the loading directions it results in pronounced asymmetrical damage induced in the longitudinal reinforcements, ultimately leading in localized damage within the RC columns.

- By employing the bond-slip model proposed in this study in finite element analysis, the load-bearing capacity of RC columns can be reproduced with high precision. When using the pull-out model, the analytical error for the maximum load capacity of Case-1 is approximately 8% to 11%. In contrast, when using the splitting model, the analytical error for the maximum load capacity of Case-2 is less than 3%. Moreover, the damage condition of the reinforcement is also replicated.

- Furthermore, pinching phenomena observed in unloading hysteresis curve of force-displacement relationship also can be reproduced by employing an appropriate bond-slip model. Thus, it enables to verify the energy dissipation during earthquakes.

- The RC column with small-diameter and high-density bar arrangements result in the splitting bond failure along the longitudinal reinforcement bars, which leads the RC column to change the failure mode from the bending deformation mode to the rocking deformation mode.

These research findings provide valuable insight into the load-bearing capacity and de-formation performance of reinforced concrete columns and the effects of bond-slip on seismic behavior. The proposed model provides an advanced prediction of the seismic

behavior of RC structures and the improvement of seismic resistance of social infrastructure facilities against earthquake disasters.

**Author Contributions:** Conceptualization, G.W., P.S. and E.E.T.; methodology, G.W. and P.S.; software, P.S.; validation, G.W. and P.S., investigation, P.S. and G.W.; resources, G.W. and P.S.; data curation, P.S.; writing—original draft preparation, P.S. and G.W.; writing—review and editing, G.W., P.S. and E.E.T.; visualization, P.S.; supervision, G.W.; funding acquisition, G.W. All authors have read and agreed to the published version of the manuscript.

**Funding:** This research received no external funding.

**Institutional Review Board Statement:** Not applicable.

**Informed Consent Statement:** Not applicable.

**Data Availability Statement:** Not applicable.

**Conflicts of Interest:** The authors declare no conflict of interest.

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
