# Peer review of "Advanced Prediction for Cyclic Bending Behavior of RC Columns Based on the Idealization of Reinforcement of Bond Properties"

_applsci, doi:10.3390/app13116379_

Round 1
Reviewer 1 Report
The study presented in the manuscript is related to the cyclic bending behavior of RC (Reinforced Concrete) columns. Two experiments with RC columns are carried out with a predefined loading protocol and different arrangement of the reinforcement, but with identical longitudinal reinforcement ratio. The experimental results are later reproduced with the DIANA FEA software, thus confirming the correctness of the FE model involved. The manuscript is well written, and its subject is original. However, the following comments are suggested to be addressed by the authors in order to improve its quality:
-Line 75: Replace "was employed to verify the numerical model and experimental results" with "was employed to verify the numerical model based on experimental results"
-Line 88: Replace "2.1. Design of Specimen with Densely Arrangement and Small Diameter Reinforcements" with "2.1. Design of Specimen with Densely Arranged, Small Diameter Reinforcements"
-Lines 149-151: Delete the two sentences written in these lines as they are repeated in the same paragraph.
-Lines 256 and 261: What does η mean in 20,000η and 4,000η respectively?
-Line 278: Please correct the phrase "In this chapter". Check in line 294 as well.
-Lines 274-276: *Calculate the critical buckling load of each rod of longitudinal reinforcement, then sum the critical loads to find the critical bending moment of the section.
-Line 285: Correct the phrase "To reduce the analysis accuracy and computation cost". Reduction of the accuracy of the analysis should not be the goal, as it is always desired to increase accuracy.
-Line 299: Replace "field of the embedded elements" with "field of the host elements"
-In Figure 17(b) replace "Modle" with "Model"
-The caption of Figure 18 seems incomplete. Please fix it.
-Increase the resolution in Figure 24(b)
-Please provide tables which show the constitutive properties of the models used for concrete material, steel material, pull-out bond failure mode and splitting bond failure mode. For example, please show the values of s1, s2 and s3 that appear in Figure 25.
-"Equivalence" of the two RC members considered in the two cases presented in the manuscript should be defined in relation to the bending response of the column, not its axial response. While in the axial response the longitudinal reinforcement ratio, either through large diameter low density bars or through small diameter high density bars, is what matters, in the bending response of the RC member the stress induced in the various reinforcement bars is not uniform; therefore the second moment of inertia of the reinforcement area or another suitable measure similar to this should be considered to declare two RC members as "equivalent". The authors could find which one of the two cases involved in the manuscript has the lower moment of inertia of the longitudinal reinforcement. The aforementioned approach could partially justify the observations of the maximum capacity drifts stated in the first conclusion, and should be stated by the authors as a limitation of the study regarding the equivalence of the two RC members involved.
-The english language of the manuscript needs some minor improvement.
Reviewer 2 Report
Most of your introduction is devoted to describing the problem and its relevance. But there are very few descriptions of similar works, their advantages and disadvantages. All analogues are described very briefly on lines 61-66. You should expand this part quite significantly. Describe the shortcomings of the approaches of other authors. Bring a greater number of works similar to yours in the review.
Lines 67-77 should probably be moved to the Methodology section. After all, in fact, this is a general plan of experimental work. It would be appropriate to bring it at the very beginning of the methodology section.
It is more convenient for the reader to present the data on lines 97-112 in the form of a table.
You write about a lot of experiences. It would be good to give a specific value, or indicate that more than xx tests were carried out on each batch of samples.
In the methodology section, I may have missed the brand of equipment on which the tests were performed. If they are not available, they must be added indicating the manufacturer. What standards were tested? This should also be stated in the article.
In general, it is necessary to work out the methodology section, indicate the equipment used, test modes, number of samples, standards, materials (in tabular form).
When describing figures 17-19, it would be good to describe in more detail where cracks originate. How reinforcement affects the initiation of cracks. In general, describe in more detail the experimental data presented in the figures.
Figure 24b, the axis labels are merged and difficult to read.
It would be at the end of section 4 to give data on how the developed model corresponds to real tests and what is the error.
The conclusions reflect the possible application of the results of the work. Also, the conclusions on lines 563-577 are descriptive. It would be nice to add numerical data and percentage change in properties.
Round 2
Reviewer 2 Report
In general, the authors have improved the article substantially. You could also add 2-3 sources on your topic to the literature review.
And in the future, try to color out what you corrected in the article. This makes it easier to search for information corrected by the authors.